# Does menopause influence the association between atherogenic index of plasma and prediabetes? A cross-sectional study in middle-aged Chinese women

Bin Ouyang[1], Hongxia Zhuo(ID)[2☯*], Huiwu Han[3,4☯*], Yujie Lei[3,4], Kangning Li[3,4], Zuxia Li[3,4]

1 Health Management Medical Center, the Third Xiangya Hospital, Central South University, Changsha, Hunan, P.R. China, 2 Hand Surgery Department, Union Hospital, Tongji Medical College, Huazhong University of Science and Technology, Wuhan, Hubei, P.R.China, 3 Teaching and Research Section of Clinical Nursing, Xiangya Hospital of Central South University, Changsha, Hunan, P.R. China, 4 Xiangya School of Nursing, Central South University, Changsha, Hunan, P.R. China

☯ These authors also contributed equally to this work.
* zhx2020zrj@163.com (HZ), hanhw8888@csu.edu.cn (HH)

**Data availability statement:** All relevant data are publicly available from the Mendeley Data repository (https://data.mendeley.com/datasets/7h7fh4k467/1).

**Funding:** HH was supported by Noncommunicable Chronic Diseases-National

## Abstract

### Background

The atherogenic index of plasma (AIP) is a novel marker associated with the risk of prediabetes, yet its interplay with menopausal status remains unclear. This study aimed to investigate the association between AIP and prediabetes among middle-aged Chinese women, and to explore this association jointly with menopausal status.

### Methods

This retrospective cross-sectional study included data of 7,929 middle-aged women who underwent physical examinations in a tertiary hospital in Changsha City, China, from 2015 to 2023. Participants' sociodemographic, physical, and laboratory data were collected from their medical records. The association between AIP and prediabetes was tested using multiple logistic regression models, followed by restricted cubic splines analysis for dose-response relationships. The relationship was further examined through stratified and joint analyses by menopausal status.

### Results

Among the 7,929 participants, 1,592 (20.08%) were newly diagnosed with prediabetes. After adjusting for confounders, AIP was associated with an increased risk of prediabetes in all women (OR: 1.45, 95% CI: 1.30, 1.62), premenopausal women (OR: 1.51, 95% CI: 1.25, 1.83), and postmenopausal women (OR: 1.44, 95% CI: 1.26, 1.65), with the relationships being approximately linear. Although the multiplicative

Science and Technology Major Project (Grant number: 2023ZD0504400), Natural Science Foundation of Hunan Province (Grant number: 2022JJ70074), Clinical Research Fund of National Clinical Research Center for Geriatric Disorders (Grant number: 2021LNJJ22), Research Project of Hunan Provincial Nursing Association (Grant number: HNKY202401). The funders had no role in study design, data collection and analysis, decision to publish, or preparation of the manuscript.

**Competing interests:** The authors have declared that no competing interests exist.

interaction was not statistically significant ($P = 0.973$), joint analysis revealed that compared to the low AIP-premenopausal reference group, the high AIP-postmenopausal group had the highest prediabetes risk (OR: 1.61, 95% CI: 1.31, 1.98).

## Conclusion

This study demonstrated a positive, linear relationship between AIP and the risk of prediabetes in middle-aged women. When considered jointly, a high AIP combined with postmenopausal status identified a subgroup with the greatest associated risk. AIP shows promise as a simple, cost-effective indicator for identifying high-risk individuals, though its clinical utility requires validation in prospective studies.

## Introduction

Prediabetes, characterized by blood glucose levels higher than normal but not meeting the diagnostic threshold for diabetes, is a highly prevalent and concerning public health issue worldwide [1]. Globally, prediabetes affects 374 million people, accounting for 7.5% of the world population. This number is projected to reach 454 million (8.0%) by 2030 and 548 million (8.6%) by 2045 [2]. China has the world's largest diabetic and prediabetic population, and a recent nationally representative survey showed that the prevalence rate of prediabetes in mainland China was as high as 35.2% [3]. Prediabetes may present insulin resistance, which, if not treated timely, may progress to diabetes [4]. In addition, prediabetes also increases the risk of developing cardiovascular diseases (CVD) [5] and microvascular complications [6]. As an intermediate state between normal blood glucose and diabetes, prediabetes can be reversed through early identification and intervention [7]. However, prediabetes remains largely underdiagnosed and undertreated, posing a significant challenge to its prevention and control [8]. Therefore, exploring risk factors for prediabetes is crucial in identifying high-risk individuals, which can help guide early prevention and interventions to delay the development of diabetes and other cardiovascular complications.

The atherogenic index of plasma (AIP) was first proposed by Dobiásová et al. [9] in 2001 as a novel biomarker of plasma atherogenicity to predict CVD risk. Several reviews and meta-analyses have consistently confirmed that a higher level of AIP was associated with increased CVD risk, greater severity, and poorer prognosis, and higher mortality across various studies with robust evidence [10–12]. In recent years, the role of AIP on prediabetes has garnered increasing research interest due to the frequently observed dyslipidemia in individuals with prediabetes [13]. Additionally, AIP has been shown to be associated with an increased risk of insulin resistance and glucose metabolism dysfunction [14], further suggesting the potential role of AIP in prediabetes. Unlike glycosylated hemoglobin which indicates established dysglycemia, AIP provides complementary pathophysiological information on the underlying metabolic disturbances that often precede hyperglycemia, making it valuable for early risk stratification. However, previous studies are predominantly

focused on the predicting effect of AIP on diabetes [15–17], and evidence on the association between AIP and prediabetes remains rare [18].

A few existing studies have explored the role of AIP in prediabetes and found that AIP was positively and non-linearly associated with the risk of prediabetes [19,20]. Additionally, a higher level of AIP at baseline has also been shown to predict an increased risk of prediabetes progression to diabetes [21,22]. Interestingly, some studies showed that the association between AIP and prediabetes was only significant in women and increased with age in subgroup analyses, suggesting that menopausal status might be a relevant factor [18,23]. Premenopausal women have high levels of estrogen, which has protective effects in inhibiting cholesterol synthesis and regulating glucose metabolism [24]. Such effects attenuate following menopause, posing a significantly increased risk of dyslipidemia and dysglycemia (diabetes or prediabetes) for postmenopausal women [25–27]. However, the nature of the association between AIP and prediabetes across different menopausal statuses, and how menopausal status and AIP may jointly relate to prediabetes risk, remain to be specifically elucidated.

Therefore, this study aimed to explore the association between AIP and prediabetes among a sample of Chinese middle-aged women, and to assess the relationship jointly by menopausal status. Our findings would offer fresh insight into the interrelationships between AIP, menopausal status, and prediabetes, which can further inform early diagnosis, prevention, and treatment for prediabetes and related complications.

## Methods

### Study design and participants

We conducted a retrospective cross-sectional study using data from a large study that aimed to explore the changes in metabolic indicators in middle-aged women. The large study collected the medical records of individuals who underwent physical examination at the Health Management Center, Xiangya Third Hospital, Central South University, Changsha City, Hunan Province from January 2015 to December 2023, the data were accessed on February 27, 2025. A total of 36,377 physical examination records were retrieved. For patients with multiple physical examinations, we only retained the most recent record. Therefore, data of 12,885 individuals were included. We excluded participants who were aged under 45 or above 59 (n = 2,944), with a diagnosis of diabetes (248), with a cognitive disorder (n = 9), with a history of cancer (n = 98), with severe liver or kidney diseases (n = 83), and whose records contained outliers or missing values (n = 1,574). Finally, 7,929 middle-aged women were included in the analysis. Fig 1 illustrates the participant exclusion process. Ethical approval was obtained from the Medical Ethics Review Committee of Xiangya Hospital, Central South University (No.: 2025020295) and informed consent was waived due to the retrospective nature of the study. The study is reported according to the Strengthening the Reporting of Observational Studies in Epidemiology (STROBE) guidelines [28] (S1 Appendix).

### Data collection

We collected participants' sociodemographic information, physical examination results, and laboratory test results from their electronic medical records and standardized questionnaires. Each category of measures was described as follows: (1) Sociodemographic information: age, education, marital status, occupation, smoking status, drinking status, exercise status, family history of diabetes, age of menarche, menopausal status, age at first childbirth, breastfeeding time, history of gestational diabetes, and history of gestational hypertension. (2) Physical examination: weight, height, waist circumference (WC), hip circumference (HC), systolic blood pressure (SBP), and diastolic blood pressure (DBP). (3) Laboratory test: fasting plasma glucose (FPG), total cholesterol (TC), triglycerides (TG), high-density lipoprotein cholesterol (HDL-C), low-density lipoprotein cholesterol (LDL-C), alanine aminotransferase (ALT), blood urea nitrogen (BUN), serum creatinine (Scr), and uric acid (UA). All blood samples were collected after an overnight fast of at least 8 hours. To ensure data quality, all research staff underwent standardized training, and a rigorous protocol of dual-person verification

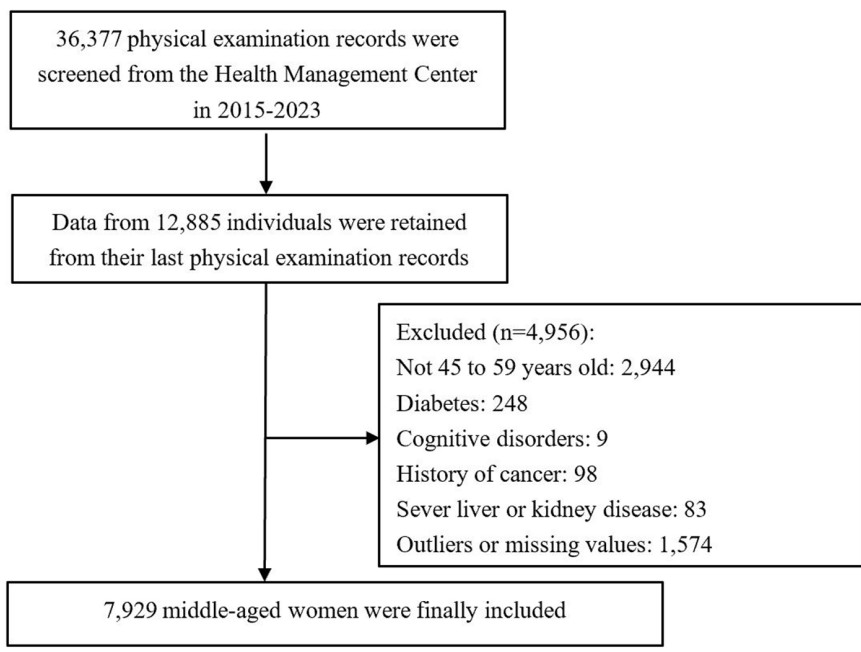

**Fig 1. Flowchart of participant selection.**

was implemented for data entry and cleaning. The exposure variable, AIP, was calculated as $\log_{10}$[(TG mmol/L)/HDL-C (mmol/L)] [9]. The outcome variable, prediabetes, was defined based on the American Diabetes Association criteria as either 5.6 mmol/L ≤ FPG < 7.0 mmol/L or self-reported prediabetes [29]. Body mass index (BMI) was calculated as weight (kg)/height$^2$(m$^2$).

## Statistical analysis

We performed statistical analysis using R software version 4.4.1 (http://www.R-project.org/), and the level of statistical significance was set at $p$-value < 0.05 for all tests. The primary analysis employed complete-case approach after assessing missing data patterns, variables with >10% missing values were excluded. Table S1 (S2 Appendix) showed the missing data proportions. Sensitivity analyses using multiple imputation validated the robustness of findings. Outliers were identified using Tukey's method (1.5 × interquartile range) and retained unless erroneous.

Participants were divided into four groups based on AIP quartiles: Q1 < −0.593, −0.593 ≤ Q2 < −0.209, −0.209 ≤ Q3 < 0.207, and Q4 > 0.207. In this study, all continuous variables were skewed and therefore expressed as median (25th to 75th interquartile range). The Kruskal-Wallis rank sum test or Wilcoxon rank sum test was thus used for comparisons between groups. Categorical variables were expressed as frequencies with percentages and compared using the chi-squared or Fisher's exact test.

We conducted multiple logistic regressions to assess the association between AIP (first treated as continuous values then categorical quartiles) and prediabetes risk in the total sample, the premenopausal group, and the postmenopausal group, respectively. We established three models for the regression analyses: The crude model did not adjust for any covariate, Model 1 was adjusted for sociodemographic information, and Model 2 was additionally adjusted for physical examinations and laboratory tests, excluding TG and HDL-C. The variance inflation factor (VIF) was used to check for multicollinearity in all logistic regression model. A restricted cubic splines (RCS) model was also constructed to explore the dose-response relationship between AIP and prediabetes, and the number of knots was selected by comparing the Akaike

Information Criterion. For the primary joint analysis, AIP was dichotomized into low and high groups based on the sample median value (−0.24). To test the robustness of our findings, we performed a sensitivity analysis in which AIP was dichotomized using a cut-off of −0.16, which has been validated in prior research [23]. Based on the AIP and menopausal status, the participants were divided into the following four groups for stratification analysis: low AIP-premenopausal group, low AIP-postmenopausal, high AIP-premenopausal, and high AIP-postmenopausal.

## Results

### Characteristics of the study participants

The participants had a median age of 52 (48.00, 54.00) years. Among the total 7,929 participants, 1,592 (20.08%) had prediabetes, and 4,763(60.07%) were currently menopausal. The number of participants in each AIP quartile was 2120 (26.74%), 2002 (25.25%), 1964 (24.77%), and 1843 (23.24%), respectively (Fig 2). Table 1 and Table S2 (S3 Appendix) show the baseline characteristics of the participants according to prediabetes status and AIP quartile, respectively. Table S3 (S4 Appendix) shows the comparison of baseline characteristics of the included and excluded participants.

### The association between AIP and prediabetes

Table 2 presents the odds ratio (OR) and 95% confidence interval (CI) for the associations between AIP and prediabetes by logistic regression analysis in the total sample and subgroups by menopausal status. VIFs for all variables in the fully adjusted model were below 10. In all three models, the continuous AIP and prediabetes showed a significant positive association, with an OR of 1.45 in the final model (95% CI: 1.30,1.62). In the AIP quartile groups, the risk of prediabetes increased with each increasing AIP quartile in all three models (*P* for trend <0.001, for all). In the fully adjusted Model 2,

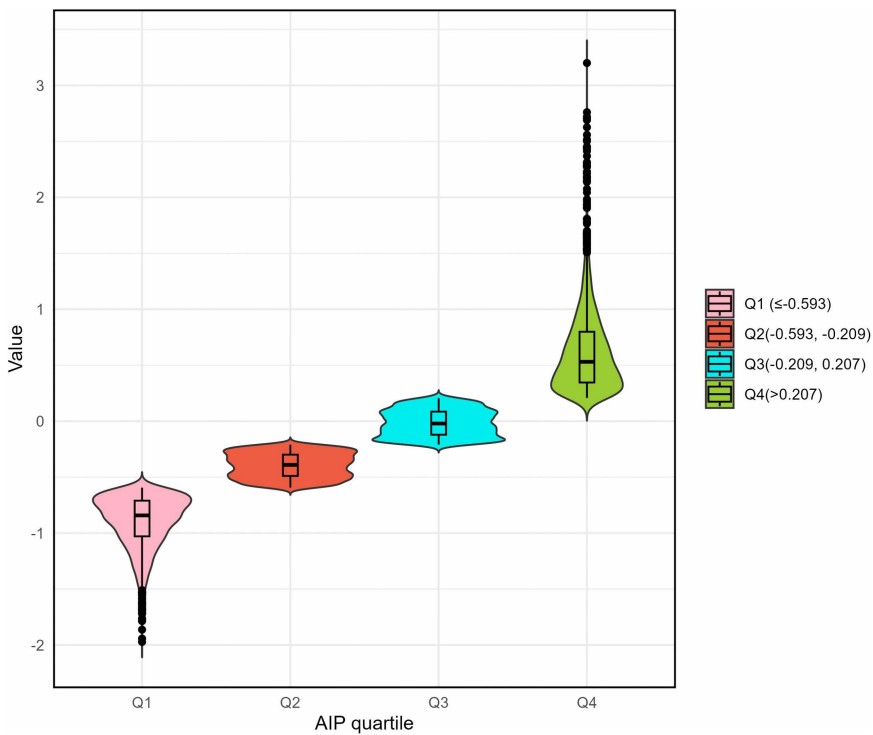

**Fig 2. Violin chart of AIP quartiles.**

**Table 1. Baseline characteristics of participants classified by prediabetes status.**

| Variables | Non-prediabetes (n = 6,337) | Prediabetes (n = 1,592) | Total (n = 7,929) | P |
|---|---|---|---|---|
| **Age (years)** | 51.00(48.00,54.00) | 52.00(49.00,55.00) | 52.00(48.00,54.00) | <0.001 |
| **Marital status** | | | | 0.553 |
| Married | 6042(95.34) | 1511(94.91) | 7553(95.26) | |
| Divorced or widowed | 251(3.96) | 66(4.15) | 317(4.00) | |
| Unmarried | 44(0.69) | 15(0.94) | 59(0.74) | |
| **Education** | | | | 0.001 |
| Junior high school or below | 515(8.13) | 147(9.23) | 662(8.35) | |
| High school | 1090(17.20) | 329(20.67) | 1419(17.90) | |
| College or above | 4732(74.67) | 1116(70.10) | 5848(73.75) | |
| **Occupation** | | | | <0.001 |
| Mental work | 3782(59.68) | 853(53.58) | 4635(58.46) | |
| Physical labor | 1702(26.86) | 464(29.15) | 2166(27.32) | |
| Unemployed | 853(13.46) | 275(17.27) | 1128(14.23) | |
| **Family history of diabetes** | | | | <0.001 |
| Yes | 950(14.99) | 303(19.03) | 1253(15.80) | |
| No | 5387(85.01) | 1289(80.97) | 6676(84.20) | |
| **Age of menarche** | | | | 0.162 |
| <12 years old | 967(15.26) | 230(14.45) | 1197(15.10) | |
| ≥12 years old | 5165(81.51) | 1296(81.41) | 6461(81.49) | |
| Not sure | 205(3.23) | 66(4.15) | 271(3.42) | |
| **Menopausal status** | | | | <0.001 |
| Premenopause | 2634(41.57) | 532(33.42) | 3166(39.93) | |
| Postmenopause | 3703(58.43) | 1060(66.58) | 4763(60.07) | |
| **Age at first childbirth** | | | | 0.358 |
| ≤20 or >35 years old | 253(3.99) | 55(3.45) | 308(3.88) | |
| 21–35 years old | 6084(96.01) | 1537(96.55) | 7621(96.12) | |
| **Breastfeeding time** | | | | 0.487 |
| <6 months | 1271(20.06) | 298(18.72) | 1569(19.79) | |
| ≥6 months | 4118(64.98) | 1051(66.02) | 5169(65.19) | |
| No breastfeeding | 948(14.96) | 243(15.26) | 1191(15.02) | |
| **Gestational diabetes** | | | | 0.212 |
| Yes | 118(1.86) | 38(2.39) | 156(1.97) | |
| No | 6219(98.14) | 1554(97.61) | 7773(98.03) | |
| **Gestational hypertension** | | | | 0.157 |
| Yes | 203(3.20) | 63(3.96) | 266(3.35) | |
| No | 6134(96.80) | 1529(96.04) | 7663(96.65) | |
| **Smoking status** | | | | 0.613 |
| Never-smoker | 6083(95.99) | 1527(95.92) | 7610(95.98) | |
| Current-smoker | 70(1.10) | 14(0.88) | 84(1.06) | |
| Ex-smoker | 12(0.19) | 5(0.31) | 17(0.21) | |
| Involuntary-smoker | 172(2.71) | 46(2.89) | 218(2.75) | |
| **Drinking status** | | | | 0.207 |
| Never-drinker | 5860(92.47) | 1471(92.40) | 7331(92.46) | |
| Current-drinker | 451(7.12) | 119(7.47) | 570(7.19) | |
| Ex-drinker | 26(0.41) | 2(0.13) | 28(0.35) | |

*(Continued)*

**Table 1.** (Continued)

| Variables | Non-prediabetes (n=6,337) | Prediabetes (n=1,592) | Total (n=7,929) | P |
|---|---|---|---|---|
| **Exercise or not** | | | | 0.601 |
| Yes | 4882(77.04) | 1216(76.38) | 6098(76.91) | |
| No | 1455(22.96) | 376(23.62) | 1831(23.09) | |
| **BMI (Kg/m²)** | 22.59(21.09,24.30) | 23.40(21.86,25.38) | 22.76(21.20,24.50) | <0.001 |
| **WC (cm)** | 76.00(72.00,80.00) | 78.00(74.00,83.00) | 76.00(72.00,81.00) | <0.001 |
| **HC (cm)** | 92.00(89.00,95.00) | 93.00(90.00,96.00) | 92.00(89.00,95.00) | <0.001 |
| **SBP (mmHg)** | 117.00(107.00,128.00) | 124.00(113.00,134.00) | 118.00(108.00,130.00) | <0.001 |
| **DBP (mmHg)** | 70.00(64.00,78.00) | 74.00(67.00,82.00) | 72.00(64.00,80.00) | <0.001 |
| **FPG (mmol/L)** | 5.09(4.83,5.32) | 5.80(5.68,5.95) | 5.21(4.91,5.51) | <0.001 |
| **TC (mmol/L)** | 5.29(4.71,5.91) | 5.44(4.80,6.13) | 5.33(4.73,5.96) | <0.001 |
| **TG (mmol/L)** | 1.17(0.87,1.61) | 1.40(1.03,1.97) | 1.21(0.89,1.68) | <0.001 |
| **HDL-C (mmol/L)** | 1.54(1.34,1.75) | 1.46(1.28,1.67) | 1.52(1.33,1.74) | <0.001 |
| **LDL-C (mmol/L)** | 3.10(2.59,3.63) | 3.16(2.64,3.75) | 3.11(2.60,3.66) | 0.001 |
| **ALT (U/L)** | 18.00(14.00,24.00) | 20.00(15.00,27.00) | 18.00(14.00,24.00) | <0.001 |
| **BUN (mmol/L)** | 4.44(3.77,5.25) | 4.60(3.91,5.43) | 4.48(3.80,5.28) | <0.001 |
| **Scr (μmol/L)** | 58.00(53.00,65.00) | 58.00(52.00,64.00) | 58.00(53.00,64.00) | 0.057 |
| **UA (μmol/L)** | 273.00(239.00,310.00) | 288.00(250.00,329.00) | 276.00(241.00,314.00) | <0.001 |
| **AIP** | −0.28(−0.66,0.11) | −0.04(−0.45,0.35) | −0.24(−0.63,0.17) | <0.001 |
| **AIP quartile** | | | | <0.001 |
| Q1 | 1840(29.04) | 280(17.59) | 2120(26.74) | |
| Q2 | 1663(26.24) | 339(21.29) | 2002(25.25) | |
| Q3 | 1530(24.14) | 434(27.26) | 1964(24.77) | |
| Q4 | 1304(20.58) | 539(33.86) | 1843(23.24) | |

AIP: atherogenic index of plasma, ALT: alanine aminotransferase, BMI: body mass index, BUN: blood urea nitrogen, DBP: diastolic blood pressure, FPG: fasting plasma glucose, HC: hip circumference, HDL-C: high-density lipoprotein cholesterol, LDL-C: low-density lipoprotein cholesterol, SBP: systolic blood pressure, Scr: serum creatinine, TC: total cholesterol, TG: triglycerides, UA: uric acid, WC: waist circumference.

compared to the lowest quartile (Q1), the adjusted Ors (95% CI) for prediabetes in the Q2, Q3, and Q4 groups were 1.20 (1.01, 1.43), 1.48 (1.24, 1.77), and 1.83 (1.53, 2.20), respectively. Similarly, in the subgroup analysis stratified by menopausal status, AIP (continuous/quartile groups) and prediabetes showed significant positive association in both premenopausal women and postmenopausal women.

## Interaction between AIP and menopausal status

To test for a potential multiplicative interaction, an interaction term (AIP × menopausal status) was included in the fully adjusted logistic regression model (Model 2). The analysis revealed no statistically significant interaction between AIP and menopausal status on the risk of prediabetes (OR for interaction = 1.00, 95% CI: 0.83, 1.22; *P* = 0.973).

## Sensitivity analysis

To assess the robustness of the association between the atherogenic index of plasma (AIP) and prediabetes, we conducted four sensitivity analyses, as summarized in Table S4 (S5 Appendix). The first analysis retained all participants without excluding those with missing data and applied multiple imputation for missing variables. The second analysis

**Table 2. The association between AIP and prediabetes by logistic regression analysis.**

| Variables | Crude model | | Model 1 | | Model 2 | |
|---|---|---|---|---|---|---|
| | OR (95% CI) | *P* | OR (95% CI) | *P* | OR (95% CI) | *P* |
| **All** | | | | | | |
| AIP | 1.81 (1.65, 1.97) | <0.001 | 1.74 (1.59, 1.90) | <0.001 | 1.45 (1.30, 1.62) | <0.001 |
| AIP quartile | | | | | | |
| Q1 | Reference | | Reference | | Reference | |
| Q2 | 1.34 (1.13, 1.59) | <0.001 | 1.31 (1.10, 1.55) | 0.002 | 1.20 (1.00, 1.43) | 0.044 |
| Q3 | 1.86 (1.58, 2.20) | <0.001 | 1.78 (1.51, 2.10) | <0.001 | 1.48 (1.24, 1.76) | <0.001 |
| Q4 | 2.72 (2.31, 3.19) | <0.001 | 2.54 (2.16, 2.99) | <0.001 | 1.83 (1.53, 2.20) | <0.001 |
| *P* for trend | | <0.001 | | <0.001 | | <0.001 |
| **Premenopause** | | | | | | |
| AIP | 1.84 (1.58, 2.14) | <0.001 | 1.81 (1.55, 2.11) | <0.001 | 1.51 (1.25, 1.83) | <0.001 |
| AIP quartile | | | | | | |
| Q1 | Reference | | Reference | | Reference | |
| Q2 | 1.55 (1.17, 2.05) | 0.002 | 1.52 (1.15, 2.02) | 0.004 | 1.40 (1.05, 1.88) | 0.023 |
| Q3 | 2.27 (1.72, 2.98) | <0.001 | 2.19 (1.66, 2.89) | <0.001 | 1.84 (1.37, 2.47) | <0.001 |
| Q4 | 2.94 (2.23, 3.88) | <0.001 | 2.83 (2.14, 3.75) | <0.001 | 1.99 (1.44, 2.74) | <0.001 |
| *P* for trend | | <0.001 | | <0.001 | | <0.001 |
| **Postmenopause** | | | | | | |
| AIP | 1.74 (1.56, 1.94) | <0.001 | 1.71 (1.52, 1.91) | <0.001 | 1.44 (1.26, 1.65) | <0.001 |
| AIP quartile | | | | | | |
| Q1 | Reference | | Reference | | Reference | |
| Q2 | 1.21 (0.97, 1.50) | 0.087 | 1.18 (0.95, 1.47) | 0.130 | 1.08 (0.86, 1.36) | 0.489 |
| Q3 | 1.62 (1.31, 1.99) | <0.001 | 1.57 (1.27, 1.94) | <0.001 | 1.32 (1.05, 1.64) | 0.015 |
| Q4 | 2.45 (2.01, 2.99) | <0.001 | 2.37 (1.94, 2.90) | <0.001 | 1.75 (1.39, 2.19) | <0.001 |
| *P* for trend | | <0.001 | | <0.001 | | <0.001 |

AIP: atherogenic index of plasma, CI: confidence interval, OR: odds ratio.

The Crude model was not adjusted for any covariate.

Model 1 was adjusted for age, education, marital status, occupation, smoking status, drinking status, exercise status, family history of diabetes, age of menarche, age at first childbirth, breastfeeding time, history of gestational diabetes, and gestational hypertension. Menopausal status was adjusted only in all participants.

Model 2 was also additionally adjusted for BMI, SBP, DBP, WC, HC, TC, LDL-C, ALT, BUN, Scr, and UA.

redefined prediabetes using a diagnostic threshold of 6.1 ≤ FPG < 7.0 mmol/L. The third analysis excluded individuals who self-reported a diagnosis of prediabetes. The fourth analysis further excluded participants using hormonal medications. Across all four analyses, AIP remained significantly and positively associated with prediabetes (all *P* < 0.001). Although the effect sizes varied somewhat across models, the overall trend was consistent, indicating that the association between AIP and prediabetes is robust under different analytical scenarios.

## Dose-response relationship

We additionally utilized RCS with 4 knots to visualize the dose-response relationship between AIP and prediabetes, and the results showed an approximately linear relationship (*p* for nonlinear = 0.080) (Fig 3A). The RCS curve suggested a general trend of increasing risk beyond an AIP of approximately −0.24. This relationship persisted in the subgroup analysis (Fig 3B).

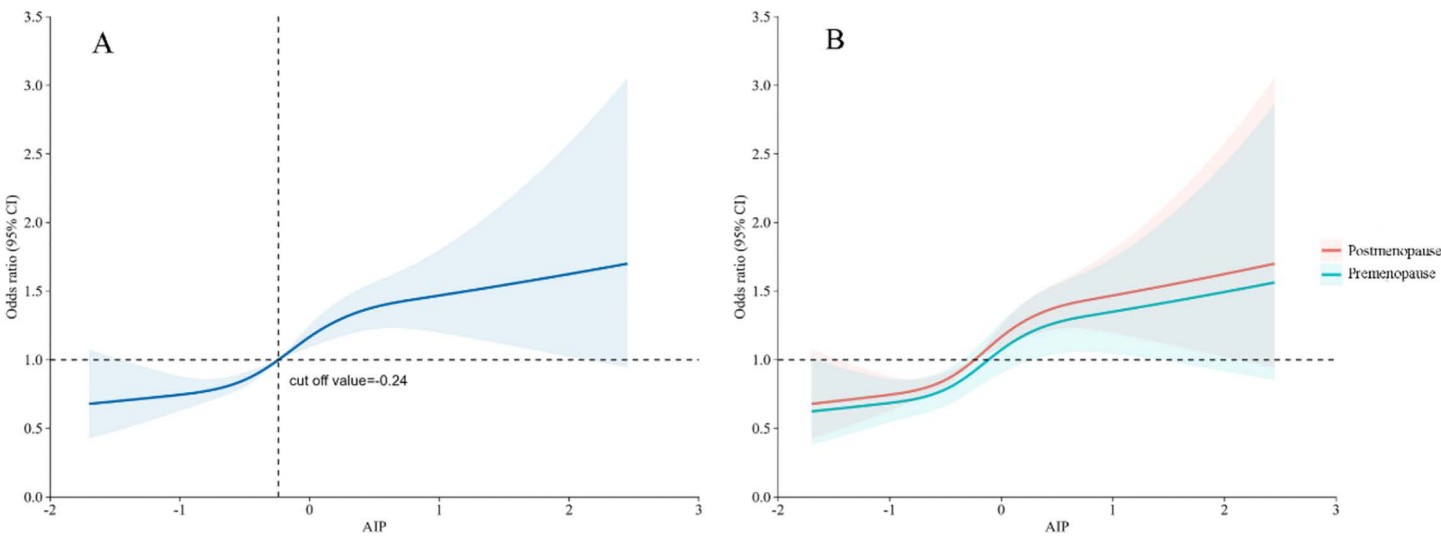

**Fig 3. The dose-response association between AIP and prediabetes.**

This model was adjusted for age, education, marital status, occupation, smoking status, drinking status, exercise status, family history of diabetes, age of menarche, age at first childbirth, breastfeeding time, history of gestational diabetes, gestational hypertension, BMI, SBP, DBP, WC, HC, TC, LDL-C, ALT, BUN, Scr, and UA.

### Joint analysis of AIP and menopausal status on prediabetes

Table 3 shows the results of joint analysis of AIP and menopause on prediabetes. Compared to the low AIP-premenopausal group, the risk of prediabetes was significantly higher in the high AIP-premenopausal (OR: 1.51, 95% CI: 1.24, 1.85) and high AIP-postmenopausal (OR: 1.61, 95% CI: 1.31, 1.98) groups. In sensitivity analysis using an AIP cut-off value of −0.16 (Table S5, S6 Appendix), the pattern of association remained consistent.

## Discussion

The association between AIP and prediabetes risk by menopausal status among middle-aged women has been rarely reported. Our study represents the first large-scale analysis to assess the interrelationships between AIP, menopausal

**Table 3. Joint analysis of AIP and menopause on prediabetes.**

| Variables | Participants | Crude model | | Model 1 | | Model 2 | |
|---|---|---|---|---|---|---|---|
| | | OR (95% CI) | *P* | OR (95% CI) | *P* | OR (95% CI) | *P* |
| Low AIP-premenopause | 1734 | Reference | | Reference | | Reference | |
| Low AIP -postmenopause | 2222 | 1.45(1.21,1.73) | <0.001 | 1.18(0.97,1.45) | 0.104 | 1.12(0.91,1.38) | 0.272 |
| High AIP-premenopause | 1432 | 2.03(1.68,2.45) | <0.001 | 1.97(1.63,2.39) | <0.001 | 1.51(1.24,1.85) | <0.001 |
| High AIP-postmenopause | 2541 | 2.62(2.21,3.09) | <0.001 | 2.09(1.72,2.54) | <0.001 | 1.61(1.31,1.98) | <0.001 |

AIP: atherogenic index of plasma, CI: confidence interval, OR: odds ratio.

The Crude model was not adjusted for any covariates.

Model 1 was adjusted for age, education, marital status, occupation, smoking status, drinking status, exercise status, family history of diabetes, age of menarche, age at first childbirth, breastfeeding time, history of gestational diabetes, and gestational hypertension.

Model 2 was also adjusted for BMI, SBP, DBP, WC, HC, TC, LDL-C, ALT, BUN, Scr, and UA based on Model 1.

status, and prediabetes risk among Chinese middle-aged women. Specifically, we used multiple analytic strategies, including logistic regression, linear regression, RCS, stratification analysis, and joint analysis to get a robust conclusion. The robustness of this association was further underscored by sensitivity analyses. Our results demonstrated that AIP was an independent risk factor for prediabetes among middle-aged women regardless of menopausal status. Notably, postmenopausal status significantly strengthens this association, with the high AIP-postmenopausal group showing the highest risk of prediabetes.

Our study identified AIP as an independent risk factor for prediabetes among middle-aged women in both multiple logistic analysis and subgroup analysis, which was consistent with previous studies [19,20]. For instance, Jiang et al.'s [19] study based on a national survey showed a significant association between AIP and the risk of prediabetes in Chinese adults aged 45 years or above. Yin et al.[30] suggest that AIP is a novel indicator that reflects the body's dyslipidemia condition, which can effectively predict abnormal glucose metabolism. AIP is a comprehensive index based on TG and HDL-C, both are easily obtainable laboratory indicators and can predict the progression from normal glucose status to prediabetes and diabetes [31,32]. Moreover, compared to traditional lipid indicators, AIP has better predictive efficacy for prediabetes [33] and IR [34] at a lower cost. Our findings highlight the potential of AIP as a cost-effective biomarker for identifying individuals at risk; however, its clinical utility as a screening tool requires further validation in prospective studies.

Interestingly, our results indicated an approximately linear association between AIP and prediabetes with a cut-off value of −0.24, which was different from previous studies showing a non-linear association [19,20]. For instance, Jiang et al.'s [19] study in a nationally representative sample showed that when AIP > 0.29, there was a positive association between AIP and the risk of prediabetes. We speculate that this discrepancy may be due to the different samples across studies. Most previous studies were conducted among the general population, but our study focused specifically on middle-aged women. Evidence suggests that the impact of AIP on prediabetes is stronger in women [29]. Therefore, AIP appears to be a risk factor for middle-aged women at a lower level compared to other studies [20,23]. These findings suggest that future studies should consider the potential gender effects when exploring the association between AIP and prediabetes.

Furthermore, we explored the association between AIP and prediabetes in relation to menopausal status. It is important to note that a formal test for a multiplicative interaction was not statistically significant, suggesting that the strength of the association may not differ fundamentally between pre- and postmenopausal women. However, we observed a numerical pattern where the association was most pronounced in postmenopausal women with high AIP levels. This pattern is biologically plausible, as menopause is accompanied with the decline in ovarian function and changes in hormonal levels, which can affect lipid and glucose metabolism [35,36]. Beyond the well-known effects on insulin secretion and sensitivity [37,38], recent advances highlight that estrogen deficiency impairs adipose tissue function, promoting a pro-inflammatory state and the release of free fatty acids, which further exacerbates insulin resistance and dysglycemia [39]. Additionally, estrogen receptors in the pancreas and liver play direct roles in beta-cell survival and hepatic glucose output, the dysfunction of which post-menopause contributes significantly to metabolic dysregulation [40]. Therefore, while AIP is a significant risk factor irrespective of menopausal status, the co-occurrence of high AIP and postmenopause status identifies a subgroup of women at particularly elevated risk. Our results suggest that the regulation and management of lipid levels should be strengthened, especially in postmenopausal women.

In addition to its large sample size, this study had several strengths. The survey questionnaire we designed for women and analyzed the impact of factors such as childbirth and menopause from multiple perspectives. Furthermore, we analyzed both the quantitative and qualitative attributes of AIP and divided middle-aged women into different subgroups.

However, some limitations of this study should be acknowledged. First, the retrospective cross-sectional design fundamentally precludes the establishment of temporality and causal inference between AIP and prediabetes, so these findings should be interpreted strictly as correlational relationships rather than evidence of causation. Second, this single-center study of middle-aged Chinese women recruited from a tertiary hospital in Changsha City may limit the generalizability

of our findings to other settings, regions, ethnicities, and age groups. Third, we lacked data on important potential confounders including insulin resistance, sex hormone levels, and detailed menopausal characteristics, which may introduce residual confounding. Fourth, some data relied on self-report and is subject to information bias. Future multi-center, prospective, longitudinal studies using more objective assessment are needed to better clarify the relationships between AIP, menopausal status, and prediabetes across other ethnic and age groups.

## Conclusion

These findings underscore the potential of AIP as a simple, cost-effective biomarker for risk stratification in middle-aged women. The results suggest that evaluating AIP, especially in combination with menopausal status, could help identify women at the highest risk, enabling more targeted monitoring. However, the cross-sectional design precludes causal inference, and the single-center setting calls for caution in generalizing the results. Future multi-center, prospective studies are essential to confirm the predictive value of AIP, establish causality, and validate the identified risk threshold across diverse ethnic and age groups.

## Supporting information

**S1 Appendix. STROBE checklist of cross-sectional study.**
(DOCX)

**S2 Appendix. Table S1 The proportion of missing variables.**
(DOCX)

**S3 Appendix. Table S2 Baseline characteristics of participants classified by quartiles of AIP.**
(DOCX)

**S4 Appendix. Table S3 Comparison of baseline characteristics of the included and excluded participants.**
(DOCX)

**S5 Appendix. Table S4 Sensitivity analysis of the association between AIP and prediabetes.**
(DOCX)

**S6 Appendix. Table S5 Sensitivity analysis for the joint association of AIP and menopause on prediabetes.**
(DOCX)

## Acknowledgments

The authors thank the engineers from the Health Management Center of Xiangya Third Hospital, Central South University, for their participation in the study.

## Author contributions

**Conceptualization:** Bin Ouyang, Hongxia Zhuo, Huiwu Han, Yujie Lei, Kangning Li, Zuxia Li.

**Data curation:** Bin Ouyang, Hongxia Zhuo, Kangning Li.

**Formal analysis:** Bin Ouyang, Yujie Lei.

**Funding acquisition:** Huiwu Han.

**Investigation:** Bin Ouyang.

**Methodology:** Hongxia Zhuo.

**Software:** Zuxia Li.

**Supervision:** Hongxia Zhuo, Huiwu Han.

**Writing – original draft:** Bin Ouyang, Hongxia Zhuo.

**Writing – review & editing:** Huiwu Han, Yujie Lei, Kangning Li, Zuxia Li.

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
