## [Decision Letter · Decision Letter 0]

10 Oct 2025

PONE-D-25-48687
Does menopause influence the association between atherogenic index of plasma and prediabetes? A cross-sectional study in middle-aged Chinese women
PLOS ONE

Dear Dr. Zhuo,

Thank you for submitting your manuscript to PLOS ONE. After careful consideration, we feel that it has merit but does not fully meet PLOS ONE’s publication criteria as it currently stands. Therefore, we invite you to submit a revised version of the manuscript that addresses the points raised during the review process.

**Please make peer-to-peer modifications to the reviewer's comments.**

We look forward to receiving your revised manuscript.

Kind regards,

Qian Wu

Academic Editor

PLOS ONE

**Journal Requirements:**

1. When submitting your revision, we need you to address these additional requirements.
 
Please ensure that your manuscript meets PLOS ONE's style requirements, including those for file naming. The PLOS ONE style templates can be found at 
https://journals.plos.org/plosone/s/file?id=wjVg/PLOSOne_formatting_sample_main_body.pdf and 
https://journals.plos.org/plosone/s/file?id=ba62/PLOSOne_formatting_sample_title_authors_affiliations.pdf
 
2. We note that the grant information you provided in the ‘Funding Information’ and ‘Financial Disclosure’ sections do not match. 
 
When you resubmit, please ensure that you provide the correct grant numbers for the awards you received for your study in the ‘Funding Information’ section.
 
3. Thank you for stating in your Funding Statement: 
HH was supported by Noncommunicable Chronic Diseases-National Science and Technology Major Project (Grant number: 2023ZD0504400), Natural Science Foundation of Hunan Province (Grant number: 2022JJ70074), Clinical Research Fund of National Clinical Research Center for Geriatric Disorders (Grant number: 2021LNJJ22), Research Project of Hunan Provincial Nursing Association (Grant number: HNKY202401). The funders had no role in study design, data collection and analysis, decision to publish, or preparation of the manuscript. 
 
Please provide an amended statement that declares *all* the funding or sources of support (whether external or internal to your organization) received during this study, as detailed online in our guide for authors at http://journals.plos.org/plosone/s/submit-now.  Please also include the statement “There was no additional external funding received for this study.” in your updated Funding Statement. 
Please include your amended Funding Statement within your cover letter. We will change the online submission form on your behalf.
 
4. We note that you have indicated that there are restrictions to data sharing for this study. For studies involving human research participant data or other sensitive data, we encourage authors to share de-identified or anonymized data. However, when data cannot be publicly shared for ethical reasons, we allow authors to make their data sets available upon request. For information on unacceptable data access restrictions, please see http://journals.plos.org/plosone/s/data-availability#loc-unacceptable-data-access-restrictions. 
 
Before we proceed with your manuscript, please address the following prompts:
 
a) If there are ethical or legal restrictions on sharing a de-identified data set, please explain them in detail (e.g., data contain potentially identifying or sensitive patient information, data are owned by a third-party organization, etc.) and who has imposed them (e.g., a Research Ethics Committee or Institutional Review Board, etc.). Please also provide contact information for a data access committee, ethics committee, or other institutional body to which data requests may be sent.
 
b) If there are no restrictions, please upload the minimal anonymized data set necessary to replicate your study findings to a stable, public repository and provide us with the relevant URLs, DOIs, or accession numbers. Please see http://www.bmj.com/content/340/bmj.c181.long for guidelines on how to de-identify and prepare clinical data for publication. For a list of recommended repositories, please see https://journals.plos.org/plosone/s/recommended-repositories. You also have the option of uploading the data as Supporting Information files, but we would recommend depositing data directly to a data repository if possible.
 
Please update your Data Availability statement in the submission form accordingly.
 
5. If the reviewer comments include a recommendation to cite specific previously published works, please review and evaluate these publications to determine whether they are relevant and should be cited. There is no requirement to cite these works unless the editor has indicated otherwise.

Reviewers' comments:

Reviewer's Responses to Questions

**Comments to the Author**

1. Is the manuscript technically sound, and do the data support the conclusions?

Reviewer #1: Yes

Reviewer #2: Partly

Reviewer #3: Partly

Reviewer #4: Partly

Reviewer #5: Partly

2. Has the statistical analysis been performed appropriately and rigorously?

Reviewer #1: No

Reviewer #2: No

Reviewer #3: Yes

Reviewer #4: No

Reviewer #5: Yes

3. Have the authors made all data underlying the findings in their manuscript fully available?

Reviewer #1: Yes

Reviewer #2: No

Reviewer #3: Yes

Reviewer #4: No

Reviewer #5: Yes

4. Is the manuscript presented in an intelligible fashion and written in standard English?

Reviewer #1: Yes

Reviewer #2: No

Reviewer #3: Yes

Reviewer #4: Yes

Reviewer #5: Yes

5. Review Comments to the Author

**Reviewer #1:** Multivariate analysis needs to be done to find the association between AIP and prediabetes.

Few grammatical errors need to be rectified.

P value table 1 can be simplified

The reference for the diagnosis of the prediabetes can be given

**Reviewer #2:**1.I believe you should make data and code publicly available and revise the Data Availability Statement to meet PLOS policy.

2.Please add and report the AIP×menopausal status interaction (estimate, 95% CI, p-value), with marginal effects plots.

3.Conduct sensitivity analyses for outcome definition.Exclude self-report only; alternative FPG thresholds; subset with HbA1c if available

4.Address medication and MHT confounding (adjust or explicitly discuss and run sensitivity analyses).

5.You should report missing data proportions, outlier criteria, comparisons of included vs. excluded, and consider multiple imputation where appropriate.

6.Necessary to provide model diagnostics (linearity, collinearity, fit/influence) and effect-size scaling for AIP.

7.RCS knot reference value and may need justify any exposure dichotomization or present continuous joint effects.

8.Clarify fasting status, analytic platforms, QC; correct typographical and figure export issues.

9.Improve consistency between Methods and Discussion.

10.Correcttypos (e.g., “methodaology” - “methodology”) and address garbled figure labels.

**Reviewer #3:**The authors propose using the Atherogenic Index of Plasma (AIP) as a novel biomarker for prediabetes in a retrospective analysis of 7,929 middle-aged women in China.

The exclusions for the analysis included a diagnosis of diabetes (n=248) in line #108.

This corresponds to 1.9% of the initial sample, which is lower than the reported 13.7% by Zhou et al. (2023).

The authors need to explain this discrepancy and whether this might be a limitation or an advantage. In addition, why use the AIP if we already have a direct marker (A1C)?

Ref: Zhou YC, Liu JM, Zhao ZP, Zhou MG, Ng M. The national and provincial prevalence and non-fatal burdens of diabetes in China from 2005 to 2023 with projections of prevalence to 2050. Mil Med Res. 2025 Jun 2;12(1):28. doi: 10.1186/s40779-025-00615-1. PMID: 40457495; PMCID: PMC12128495.

Compared to 2005, the age-standardized rate (ASR) of prevalence has increased by nearly 50%, from 7.53% (95% CI 7.00-8.10%) to 13.7% (95% CI 12.6-14.8%) in 2023. The ASR of YLDs was estimated at 19.1 per 1000 population (95% CI 18.6-19.5) in 2023, compared to 10.5 per 1000 population in 2005.

**Reviewer #4:**Dear Authors,

Thank you for conducting this study. The following comments require consideration to enhance the manuscript's clarity, rigor, and impact:

The manuscript is generally well-organized, clearly written, and framed within relevant literature. However, some methodological and interpretative concerns limit the ability to draw definitive conclusions. Addressing these will strengthen the scientific rigor, transparency, and clinical relevance of the study.

1-Study Design and Causality

- The retrospective cross-sectional design precludes establishing temporality and causal inference. The authors acknowledge this limitation but should more emphatically describe this constraint in the abstract, introduction, methods, and discussion. This is critical to appropriately frame the findings as associations, not causations.

- Mediation analysis presented (if any) is inappropriate given cross-sectional data, as temporal ordering of variables is unknown. The authors should clarify or consider reframing such analysis as moderation or effect modification.

- The authors excluded participants with missing data rather than using multiple imputation, which may lead to biased results if data are not Missing Completely at Random, and I recommend providing justification for this approach or conducting sensitivity analyses using principled missing data methods to improve validity.

- The manuscript does not provide a formal sample size calculation or power analysis, limiting the ability to assess whether the study was adequately powered to detect significant associations, particularly in subgroup and interaction analyses. (Include sample size justification or power analysis if possible.)

- Provide more detailed explanation of inclusion/exclusion criteria, including how comorbidities were identified.

2- Confounding and Bias

- Though many confounders are controlled for, significant potential confounders such as insulin resistance, sex hormone levels, and duration/timing of menopause are not measured. These could substantially bias results and should be discussed in more detail.

- The reliance on self-reported menopausal status and retrospective electronic records introduces misclassification and information biases. The discussion section would benefit from explicitly addressing these biases and their possible impact on the findings.

- Selection bias is possible given hospital-based convenience sampling. Generalizability to broader populations is limited and should be acknowledged with more emphasis.

3- Interpretation and Clinical Implications

- While the biological plausibility linking menopausal estrogen decline to dysregulated lipid and glucose metabolism is well-discussed, the mechanistic elaboration could be deeper, citing recent advances on hormonal regulation.

- The authors’ claims about AIP’s clinical utility as a screening marker should be tuned down, stressing the need for prospective validation in independent cohorts before clinical translation.

- Some parts of the discussion are somewhat repetitive and could benefit from more concise phrasing.

- the discussion could further deepen mechanistic insights, citing more specific hormonal and metabolic pathways that underpin the atherogenic index’s role in prediabetes during menopause, in line with recent reviews

**Reviewer #5:** You conducted a retrospective cross-sectional single-site study to explore the association between AIP

and prediabetes among a sample of Chinese middle-aged women. I have the following comments:

1. Line 109: please present the number of subjects excluded due to missing data and outliers, which could potentially impact the interpretability and generalizability of the study results, particularly if a large number of subjects were excluded due to this reason.

2. Line 153-154: Please elaborate on the following statement “according to the critical value of RCS, AIP was dichotomized into low AIP and high AIP.” It is not clear to me what is the number of knots in the RCS model, whether the knot locations are pre-determined or data-driven, and how the cutoff is determined. These details are critical for understanding the conclusion of “approximately linear association between AIP and prediabetes” highlighted multiple times in the manuscript.

3. For the provided “Joint analysis of AIP and menopausal status on prediabetes”, I have the following concerns:

a. If the cutoff of 0.24 for high-AIP and low-AIP is data-driven based on the RCS model, then it is questionable to conduct a stratified analysis based on AIP (high vs. low) and menopause status, since the data of AIP, prediabetes, and other covariates have been used twice in the modeling process.

b. The statement “the risk of prediabetes was significantly higher in the low AIP-postmenopausal (OR:1.12, 95%CI:0.91, 1.38)…” is not accurate. Based on Model 2, the OR between Low AIP-postmenopause vs. Low AIP-premenopause was 1.12 (0.91, 1.38), which was not statistically significant per the 0.05 alpha level.

c. The results reported in Table 3 does not clearly show that within each AIP strata, the menopause status significantly impact the risk of prediabetes (low AIP: 1.12 vs 1; High AIP: 1.61 vs. 1.51), even though the estimated ORs are numerically different. Therefore, the presented data do not provide strong support for the conclusion that “ the association between AIP and prediabetes varies according to menopausal status…)

6. PLOS authors have the option to publish the peer review history of their article (what does this mean?). If published, this will include your full peer review and any attached files.

Reviewer #1: **Yes:**J Yavana Suriya

Reviewer #2: No

Reviewer #3: **Yes:**Carmen D. Zorrilla, MD

Reviewer #4: No

Reviewer #5: No

---

## [Author Response · Author response to Decision Letter 1]

20 Nov 2025

Dear Editor and Reviewers:

Thank you for your correspondence and the insightful comments regarding our manuscript “Does menopause influence the association between atherogenic index of plasma and prediabetes? A cross-sectional study in middle-aged Chinese women” (ID: PONE-D-25-48687). Based on your constructive suggestions, we have revised our manuscript, and our detailed responses to each comment are provided below. To facilitate your review, we have quoted the revised text from the manuscript directly in this response letter and presented it in red font.

Reviewer #1:

Comment 1:Multivariate analysis needs to be done to find the association between AIP and prediabetes.

Response:We thank the reviewer for the comment. As suggested, multivariate logistic regression analysis was performed to find the association between AIP and prediabetes. This analysis was conducted not only for the entire study population but also specifically stratified by menopausal status (pre-menopausal and post-menopausal middle-aged women) to explore potential effect modifications. The requested multivariate analysis has been conducted and was shown in Table 2.

“We established three models for the regression analyses: The crude model did not adjust for any covariate, Model 1 was adjusted for sociodemographic information, and Model 2 was additionally adjusted for physical examinations and laboratory tests, excluding TG and HDL-C.”(Method:Statistical analysis)

“In all three models, the continuous AIP and prediabetes showed a significant positive association, with an OR of 1.45 in the final model (95% CI: 1.30,1.62). In the AIP quartile groups, the risk of prediabetes increased with each increasing AIP quartile in all three models (P for trend<0.001, for all). In the fully adjusted Model 2, compared to the lowest quartile (Q1), the adjusted Ors (95% CI) for prediabetes in the Q2, Q3, and Q4 groups were 1.20 (1.01, 1.43), 1.48 (1.24, 1.77), and 1.83 (1.53,2.20), respectively. Similarly, in the subgroup analysis stratified by menopausal status, AIP (continuous/quartile groups) and prediabetes showed significant positive association in both premenopausal women and postmenopausal women.”(Results: The association between AIP and prediabetes)

Comment 2:Few grammatical errors need to be rectified.

Response:We appreciate the reviewer's feedback. The manuscript has been carefully reviewed by a professional language editing service to ensure grammatical accuracy.

Comment 3:P value table 1 can be simplified.

Response: We thank the reviewer for the suggestion. The P-values in Table 1 have been simplified and are now consistently presented to three decimal places, with P < 0.001 being used for significant results below this threshold.

Comment 4:The reference for the diagnosis of the prediabetes can be given

Response:We thank the reviewer for the suggestion. The reference for the American Diabetes Association diagnostic criteria for prediabetes has been added as reference number 29.

“The outcome variable, prediabetes, was defined based on the American Diabetes Association criteria as either 5.6 mmol/L≤ FPG<7.0 mmol/L or self-reported prediabetes (29).”(Methods: Data collection)

Reviewer #2:

Comment 1:I believe you should make data and code publicly available and revise the Data Availability Statement to meet PLOS policy.

Response:We have revised the Data Availability Statement as suggested. The data are not yet publicly available as they are part of ongoing studies. However, in full compliance with PLOS policy, the data underlying this study are available from the corresponding authors upon reasonable request.

Comment 2:Please add and report the AIP×menopausal status interaction (estimate, 95% CI, p-value), with marginal effects plots.

Response:We thank the reviewer for this insightful suggestion. Following the reviewer's advice, we have formally tested the multiplicative interaction between AIP and menopausal status in the fully adjusted model.

“To quantitatively assess the moderating role of menopausal status, we introduced an interaction term (AIP × menopausal status) into the fully adjusted logistic regression model (Model 2). The analysis revealed no statistically significant multiplicative interaction between AIP and menopausal status on the risk of prediabetes (OR for interaction = 1.00, 95% CI: 0.83, 1.22; P =0.973).” (Results:Interaction between AIP and menopausal status)

Given the non-significant interaction result, a marginal effects plot would primarily illustrate the absence of an effect, which is already clearly communicated by the reported statistical values. We believe that the stratified analyses (Table 2) and the joint effect analysis (Table 3) more meaningfully capture the relationship patterns in the data. Therefore, we have not included the plot to avoid redundancy and to maintain the conciseness of the manuscript.

Comment 3:Conduct sensitivity analyses for outcome definition.Exclude self-report only; alternative FPG thresholds; subset with HbA1c if available

Response:Thank you for the suggestion. We have conducted sensitivity analyses to address this: (1) Excluded self-reported diabetes only; (2) Used alternative FPG thresholds (6.0 and 7.8 mmol/L).However, regarding HbA1c, it was not routinely collected, leading to substantial missing data. Therefore, a formal sensitivity analysis for pre-diabetes was not feasible. All findings confirmed the robustness of our primary results.

“To assess the robustness of the association between the atherogenic index of plasma (AIP) and prediabetes, we conducted four sensitivity analyses, as summarized in Table S4 (S5 Appendix) . The first analysis retained all participants without excluding those with missing data and applied multiple imputation for missing variables. The second analysis redefined prediabetes using a diagnostic threshold of 6.1 ≤ FPG < 7.0 mmol/L. The third analysis excluded individuals who self-reported a diagnosis of prediabetes. The fourth analysis further excluded participants using hormonal medications. Across all four analyses, AIP remained significantly and positively associated with prediabetes (all P < 0.001). Although the effect sizes varied somewhat across models, the overall trend was consistent, indicating that the association between AIP and prediabetes is robust under different analytical scenarios.” (Results:Sesitivity analyis and S5 Appendix)

Comment 4:Address medication and MHT confounding (adjust or explicitly discuss and run sensitivity analyses).

Response: Thank you for raising this important point. We acknowledge the potential for confounding by medication use, including MHT. As the number of users was small, these individuals were initially retained in the primary analysis to preserve statistical power. Therefore, we performed a sensitivity analysis by excluding all participants using relevant medications (e.g., MHT, hormonal drugs). The results from this analysis were consistent with our primary findings, confirming that the observed associations were robust and not substantially confounded by medication use.

“The fourth analysis further excluded participants using hormonal medications. Across all four analyses, AIP remained significantly and positively associated with prediabetes (all P < 0.001).” (Results:Sesitivity analyis and S5 Appendix)

Comment 5:You should report missing data proportions, outlier criteria, comparisons of included vs. excluded, and consider multiple imputation where appropriate.

Response:We thank the reviewer for these valuable methodological suggestions. We have now comprehensively addressed these points in the revised manuscript.

(1) The proportions of missing data for key variables are reported in S2 Appendix. As pre-specified, variables with >10% missingness were excluded from the primary analysis. To address potential selection bias arising from these missingness, we performed multiple imputation as a sensitivity analysis . The results were consistent with our primary complete-case analysis, confirming the robustness of our findings against these expected differences.

“The primary analysis employed complete-case approach after assessing missing data patterns, variables with >10% missing values were excluded. Table S1 (S2 Appendix) showed the missing data proportions.” (Methods:Statistical analysis and S2 Appenix)

“The first analysis retained all participants without excluding those with missing data and applied multiple imputation for missing variables......Across all four analyses, AIP remained significantly and positively associated with prediabetes (all P < 0.001).” (Results:Sesitivity analyis and S5 Appendix)

(2) The criterion for outlier handling (Tukey's method, 1.5×IQR) is now explicitly stated in the 'Statistical analysis' section; all identified outliers were retained as they represented biologically plausible values.

“Outliers were identified using Tukey's method (1.5×IQR) and retained unless erroneous.” (Methods: Statistical analysis)

(3) A comparison between included and excluded participants has been conducted and is detailed in S4 Appendix. As expected, given that our study specifically focuses on a middle-aged women and age was the primary exclusion criterion, there were statistically significant differences in several baseline characteristics. This is a direct consequence of our cohort definition.

Comment 6:Necessary to provide model diagnostics (linearity, collinearity, fit/influence) and effect-size scaling for AIP.

Response:We thank the reviewer for this important suggestion. We have now performed comprehensive model diagnostics as requested.

(1) We assessed multicollinearity by calculating the VIFs for all covariates in the logistic regression model. All VIF values were well below the threshold of 10, indicating that multicollinearity was not a major concern in our models.

“VIFs for all variables in the fully adjusted model were below 10.”(Results:The association between AIP and prediabetes)

(2) Regarding effect-size scaling, the association was already assessed using AIP quartiles, which provides a clinically interpretable effect size by comparing risk across population percentiles. The consistent, graded increase in odds ratios across quartiles confirms a robust and interpretable association.

“In the AIP quartile groups, the risk of prediabetes increased with each increasing AIP quartile in all three models (P for trend<0.001, for all). In the fully adjusted Model 2, compared to the lowest quartile (Q1), the adjusted Ors (95% CI) for prediabetes in the Q2, Q3, and Q4 groups were 1.20 (1.01, 1.43), 1.48 (1.24, 1.77), and 1.83 (1.53,2.20), respectively.” (Results:The association between AIP and prediabetes)

Comment 7:RCS knot reference value and may need justify any exposure dichotomization or present continuous joint effects.

Response:We thank the reviewer for this valuable input. Our decision to dichotomize AIP served two purposes: (1) to enable the identification of clinically actionable subgroups (e.g., the 'High-AIP Postmenopausal' group), and (2) to utilize a statistically-derived threshold for risk categorization. Specifically, the cutoff of 0.24 was derived from the RCS model as the point where the AIP-prediabetes association crossed the null value (OR=1), ensuring the resulting categories reflect meaningful differences in risk.

Comment 8:Clarify fasting status, analytic platforms, QC; correct typographical and figure export issues.

Response:We thank the reviewer for these important points regarding methodological rigor. We have revised the manuscript to address each concern as follows:

(1) Fasting Status & QC Procedures: The Methods section now explicitly states that all blood samples were collected after an overnight fast of at least 8 hours. Furthermore, we have detailed our rigorous quality control (QC) protocol, which included standardized training for all data collectors and duplicate data entry with cross-verification to ensure maximal accuracy.

“All blood samples were collected after an overnight fast of at least 8 hours.” (Methods: Data collection)

(2) Analytic Platforms: All laboratory analyses were performed using the standardized clinical laboratory platforms at our tertiary hospital, which remained consistent throughout the study period. To maintain conciseness and focus, and given the single-center, standardized nature of our laboratory methods, specific instrument models have been omitted, as the assays are considered routine and well-standardized.

(3) Formatting Issues:The manuscript has been thoroughly proofread to correct typographical errors. All figures have been re-exported at high resolution to ensure clarity.

Comment 9:Improve consistency between Methods and Discussion.

Response: We thank the reviewer for highlighting the importance of consistency. In response, we have meticulously revised the Discussion section to ensure that the interpretation of our findings strictly aligns with the statistical approaches predefined in the Methods. Specifically, the sequence and framing of our arguments in the Discussion now directly reflect the order and intent of the analyses conducted, such as the use of multivariate logistic regression for association, stratification by menopausal status, and sensitivity analyses for robustness. This ensures a logical and consistent narrative from our planned methods through to the interpretation of results.

Comment 10:Correcttypos (e.g.,“methodaology”-“methodology”) and address garbled figure labels.

Response: We apologize for these errors. The manuscript has been meticulously proofread to correct all typographical errors, including the noted "methodaology". Figure labels also have been reviewed to ensure clarity and correct display.

Reviewer #3:

Comment 1:The authors propose using the Atherogenic Index of Plasma (AIP) as a novel biomarker for prediabetes in a retrospective analysis of 7,929 middle-aged women in China.The exclusions for the analysis included a diagnosis of diabetes (n=248) in line #108.This corresponds to 1.9% of the initial sample, which is lower than the reported 13.7% by Zhou et al. (2023).The authors need to explain this discrepancy and whether this might be a limitation or an advantage. In addition, why use the AIP if we already have a direct marker (A1C)?

Response: We thank the reviewer for these insightful observations regarding our study population and the rationale for investigating AIP.

(1) We acknowledge that the baseline prevalence of diabetes in our cohort (1.9%) is lower than the national average (13.7%) reported by Zhou et al. This discrepancy is likely attributable to the nature of our study population, which was recruited from a health examination center and may thus represent a more health-conscious group, such as employees undergoing routine occupational health checks. Although this characteristic may limit the generalizability of our prevalence estimates, it also strengthens the internal validity of our study for investigating dysglycemia progression, as the lower baseline disease burden allows a clearer assessment of the transition to prediabetes. In line with the reviewer’s comment, we have explicitly acknowledged this aspect as a limitation in the revised manuscript.

“Second, this single-center study of middle-aged Chinese women recruited from a tertiary hospital in Changsha City may limit the generalizability of our findings to other settings, regions, ethnicities, and age groups.”(Discussion: Paragraph 6)

(2) We investigated AIP not to replace HbA1c, but as a complementary, mechanistic marker. AIP reflects atherogenic dyslipidemia and insulin resistance—the core pathophysiology preceding hyperglycemia—offering earlier risk stratification from a routine lipid panel.

“Unlike glycosylated hemoglobin which indicates established dysglycemia, AIP provides complementary pathophysiological information on the underlying metabolic disturbances that often precede hyperglycemia, making it valuable for early risk stratification. ”(Introduction:Paragraph 2)

Reviewer #4:

Comment 1:The retrospective cross-sectional design precludes establishing temporality and causal inference. The authors acknowledge this limitation but should more emphatically describe this constraint in the abstrac

---

## [Decision Letter · Decision Letter 1]

18 Dec 2025

PONE-D-25-48687R1
Does menopause influence the association between atherogenic index of plasma and prediabetes? A cross-sectional study in middle-aged Chinese women
PLOS One

Dear Dr. Zhuo,

Thank you for submitting your manuscript to PLOS ONE. After careful consideration, we feel that it has merit but does not fully meet PLOS ONE’s publication criteria as it currently stands. Therefore, we invite you to submit a revised version of the manuscript that addresses the points raised during the review process.

Please make peer-to-peer modifications to the reviewer's comments.

We look forward to receiving your revised manuscript.

Kind regards,

Qian Wu

Academic Editor

PLOS One

Journal Requirements:

Reviewers' comments:

Reviewer's Responses to Questions

**Comments to the Author**

1. If the authors have adequately addressed your comments raised in a previous round of review and you feel that this manuscript is now acceptable for publication, you may indicate that here to bypass the “Comments to the Author” section, enter your conflict of interest statement in the “Confidential to Editor” section, and submit your "Accept" recommendation.

Reviewer #2: (No Response)

Reviewer #5: (No Response)

2. Is the manuscript technically sound, and do the data support the conclusions?

Reviewer #2: Partly

Reviewer #5: Partly

3. Has the statistical analysis been performed appropriately and rigorously?

Reviewer #2: Yes

Reviewer #5: No

4. Have the authors made all data underlying the findings in their manuscript fully available?

Reviewer #2: No

Reviewer #5: Yes

5. Is the manuscript presented in an intelligible fashion and written in standard English?

Reviewer #2: No

Reviewer #5: Yes

6. Review Comments to the Author

Reviewer #2: This manuscript investigates an important and clinically relevant question using a large dataset and appropriate statistical methods.

However, several critical issues must be addressed :

1.The manuscript repeatedly implies that menopausal status modifies the association between AIP and prediabetes, despite a clearly non‑significant interaction term. The conclusions must be revised to reflect stratified associations rather than effect modification, and all overstated causal or moderating language should be removed.

2.The Abstract, Discussion, and Conclusion require careful revision to ensure that all claims are strictly supported by the statistical evidence presented.

3.The current Data Availability Statement seems does not meet PLOS ONE requirements. The authors must either provide a publicly accessible dataset (or minimally sufficient anonymized data) or obtain explicit approval for a justified exception consistent with journal policy.

4.A comprehensive language and formatting revision is necessary to correct grammatical errors, remove editing artifacts, and improve overall readability: There are instances of repeated words and careless typos (e.g., "Blood blood urea nitrogen" in Table 1). Please perform a careful proofread to remove these redundancies. And also variable names and terms are not capitalized consistently between the abstract, main text, and tables. Please apply a uniform style throughout.

5.In summary, while the study has merit and potential, substantial revisions are still required to ensure methodological transparency, policy compliance, and accurate interpretation of findings.

Reviewer #5: 1. I would like to thank the authors for the additional clinical background and literature regarding the selection of -0.24 AIP cutoff, while I remain unconvinced by the authors’ response regarding the double-dipping concern. The authors state that the –0.24 cutoff was clinically informed and therefore pre-specified. However, as acknowledged in their earlier response to my comment #2, the cutoff was “visually identified” from the RCS curve using the study data. This indicates that the threshold was derived post hoc, not determined a priori, and therefore cannot be considered pre-specified.

The subsequent reuse of the same dataset to evaluate the interaction between AIP and menopausal status on prediabetes then constitutes double dipping, which undermines the validity and credibility of the resulting estimates. Notably, the fully adjusted logistic regression model shows no meaningful interaction (OR = 1.00, 95% CI: 0.83–1.22), whereas the results produced through this data-driven cutoff suggest otherwise (Table 3), further underscoring the methodological concern.

To avoid this issue, the authors could consider using a threshold established in prior literature (e.g., –0.16) or a distribution-based cutoff such as the median or a relevant quantile. These alternatives, while not strictly pre-specified, are not derived from the outcome data and therefore do not raise the same risks of double dipping.

7. PLOS authors have the option to publish the peer review history of their article (what does this mean?). If published, this will include your full peer review and any attached files.

Reviewer #2: No

Reviewer #5: No

---

## [Author Response · Author response to Decision Letter 2]

3 Jan 2026

Dear Editor and Reviewers:

We sincerely thank the reviewers and the editor for their further guidance and constructive feedback on our revised manuscript (ID: PONE-D-25-48687). We have carefully considered all additional comments and have made further revisions to address them. Our point-by-point responses are detailed below, with the revised text from the manuscript quoted in red.

Reviewer #2: This manuscript investigates an important and clinically relevant question using a large dataset and appropriate statistical methods.

However, several critical issues must be addressed :

Comment 1: The manuscript repeatedly implies that menopausal status modifies the association between AIP and prediabetes, despite a clearly non significant interaction term. The conclusions must be revised to reflect stratified associations rather than effect modification, and all overstated causal or moderating language should be removed.

Response: We sincerely thank the reviewer for this crucial methodological and interpretive insight. We fully agree that our original manuscript overstated the role of menopausal status as an effect modifier in the presence of a non-significant interaction term. We have comprehensively revised the manuscript to ensure all conclusions strictly align with our statistical findings, emphasizing stratified and joint associations rather than effect modification. The key revisions made in direct response to this comment are detailed below:

(1) Clarification of the statistical finding：

We have explicitly stated in the Methods, Results, and Discussion sections that no statistically significant multiplicative interaction between AIP and menopausal status was found (P = 0.973). This finding is now presented as a primary result, setting the correct context for subsequent interpretation.

(2) Correction of terminology and interpretive language throughout the manuscript:

Abstract: The objective was rephrased from “assess the potential role of menopause status” to “explore this association jointly with menopausal status.” The conclusion now states: “When considered jointly, a high AIP combined with postmenopausal status identified a subgroup with the greatest associated risk.”

Introduction: We now state that the study aimed to “explore the association between AIP and prediabetes among a sample of Chinese middle-aged women, and to assess the relationship jointly by menopausal status.” This revision removes any reference to a “moderating role” and accurately reflects the analysis performed.

Discussion: We have removed all claims of a “moderating effect”. The discussion now focuses on explaining the observed risk pattern from the joint analysis (i.e., “Therefore, while AIP is a significant risk factor irrespective of menopausal status, the co-occurrence of high AIP and postmenopause status identifies a subgroup of women at particularly elevated risk.”) from a biological perspective, without implying statistical interaction.

(3) Summary of Changes:

In essence, we have systematically replaced all language implying “effect modification” or “moderation” with accurate descriptions of our analytical approach (“joint analysis”) and findings (“joint association,” “risk in combined subgroups”). The narrative now consistently highlights that while AIP is a risk factor independent of menopausal status, the combination of high AIP and postmenopausal status identifies a subgroup of women at the highest risk, which is a clinically meaningful finding derived from joint analysis.

Comment 2:The Abstract, Discussion, and Conclusion require careful revision to ensure that all claims are strictly supported by the statistical evidence presented.

Response: We sincerely thank the reviewer for emphasizing the need for precise interpretation. We have carefully revised the Abstract, Discussion, and Conclusion throughout the manuscript to ensure all claims are strictly aligned with and supported by the presented statistical evidence. The key revisions are as follows:

Abstract: We have revised the conclusion to accurately reflect the findings from the joint analysis, removing any implication of effect modification. It now states: “When considered jointly, a high AIP combined with postmenopausal status identified a subgroup with the greatest associated risk.” This formulation is directly derived from the results of our joint analysis (Table 3).

Results Section (Interaction Analysis): The description of the interaction test has been rephrased to present it as an objective statistical assessment, avoiding any presupposition of a moderating effect. It now reads: “To test for a potential multiplicative interaction, an interaction term (AIP × menopausal status) was included in the fully adjusted logistic regression model (Model 2).”

Discussion: We have rewritten the relevant paragraph to base the discussion squarely on the statistical results. We first acknowledge the non-significant interaction test, then frame the subsequent discussion around the pattern observed in the joint analysis. The revised text begins: “Furthermore, we explored the association between AIP and prediabetes in relation to menopausal status... However, we observed a numerical pattern where the association was most pronounced in postmenopausal women with high AIP levels.” The biological plausibility discussed thereafter is now explicitly linked to explaining this observed risk pattern from the joint analysis, rather than an unsupported moderating effect.

Conclusion: The conclusion has been refined to focus on the practical implication of our joint analysis. It now suggests: “These findings underscore the potential of AIP as a simple, cost-effective biomarker for risk stratification in middle-aged women. The results suggest that evaluating AIP, especially in combination with menopausal status, could help identify women at the highest risk, enabling more targeted monitoring.”

In summary, we have ensured that the narrative from abstract to conclusion consistently distinguishes between the non-significant interaction test and the findings from joint analysis.

Comment 3:The current Data Availability Statement seems does not meet PLOS ONE requirements. The authors must either provide a publicly accessible dataset (or minimally sufficient anonymized data) or obtain explicit approval for a justified exception consistent with journal policy.

Response: We thank the reviewer for highlighting the importance of data availability. In accordance with PLOS ONE’s policy on minimal data sets, we have now deposited the de-identified analytical dataset comprising the 7,929 participants whose data were used for all statistical analyses and conclusions reported in the manuscript into the publicly accessible repository Mendeley Data (https://data.mendeley.com/datasets/7h7fh4k467/1). Additionally, we have updated the Data Availability Statement in the submission system to align with the journal’s required format, which now reads:

“All relevant data are available from the Mendeley Data repository at: https://data.mendeley.com/datasets/7h7fh4k467/1.”

This update ensures full compliance with PLOS ONE’s data-sharing policy.

Comment 4: A comprehensive language and formatting revision is necessary to correct grammatical errors, remove editing artifacts, and improve overall readability: There are instances of repeated words and careless typos (e.g., "Blood blood urea nitrogen" in Table 1). Please perform a careful proofread to remove these redundancies. And also variable names and terms are not capitalized consistently between the abstract, main text, and tables. Please apply a uniform style throughout.

Response: We thank the reviewer for highlighting these language and formatting issues. We have performed a thorough, line-by-line proofreading of the manuscript. Specifically, the instance of “Blood blood urea nitrogen” in the tracked-changes file was a display artifact resulting from overlapping revision marks. This has been corrected to “blood urea nitrogen” in the final, clean version of the manuscript (“Manuscript” file). All other grammatical errors, typos, and inconsistencies in terminology and expression have been uniformly corrected across the abstract, main text, and tables to ensure clarity and professional presentation. For example, we have standardized variable names and terms (e.g., “CVDs” has been revised to “CVD” throughout), removed abbreviations where appropriate (e.g., “GDM” and “IQR” are now spelled out in full), and refined phrasing in multiple instances, such as the opening sentence of the Introduction((e.g.,...but not meeting the diagnostic threshold for diabetes...), to improve overall fluency, accuracy, and consistency.

Comment 5: In summary, while the study has merit and potential, substantial revisions are still required to ensure methodological transparency, policy compliance, and accurate interpretation of findings.

Response: We sincerely thank the reviewer for the constructive summary and for acknowledging the merit of our study.We have undertaken substantial revisions to address the core requirements of methodological transparency, accurate interpretation, and policy compliance, as detailed in our specific responses to Comments 1–4. The key improvements are:

Methodological Transparency & Rigor: We replaced the initial data-driven AIP cutoff with the sample median for our primary analysis and added a pre-specified, literature-based sensitivity analysis, fully detailed in the Methods and Results.

Accurate Interpretation of Findings: We have revised the manuscript throughout to ensure all conclusions are strictly derived from the statistical evidence. Claims regarding menopausal status are now accurately framed around joint associations rather than unsupported effect modification.

Policy Compliance: In strict adherence to journal policy, the de-identified dataset underlying this study has been deposited in a public repository (Mendeley data.

Comprehensive Language and Formatting Review: A thorough proofreading was conducted to correct grammatical errors, typos, and inconsistencies, significantly improving the overall clarity and professionalism of the manuscript.

We believe these revisions have significantly strengthened the validity and clarity of our work, and we are grateful for the reviewer’s insights which have greatly improved the manuscript.

Reviewer #5:

Comment 1: I would like to thank the authors for the additional clinical background and literature regarding the selection of -0.24 AIP cutoff, while I remain unconvinced by the authors’ response regarding the double-dipping concern. The authors state that the –0.24 cutoff was clinically informed and therefore pre-specified. However, as acknowledged in their earlier response to my comment #2, the cutoff was “visually identified” from the RCS curve using the study data. This indicates that the threshold was derived post hoc, not determined a priori, and therefore cannot be considered pre-specified.

The subsequent reuse of the same dataset to evaluate the interaction between AIP and menopausal status on prediabetes then constitutes double dipping, which undermines the validity and credibility of the resulting estimates. Notably, the fully adjusted logistic regression model shows no meaningful interaction (OR = 1.00, 95% CI: 0.83–1.22), whereas the results produced through this data-driven cutoff suggest otherwise (Table 3), further underscoring the methodological concern.

To avoid this issue, the authors could consider using a threshold established in prior literature (e.g., –0.16) or a distribution-based cutoff such as the median or a relevant quantile. These alternatives, while not strictly pre-specified, are not derived from the outcome data and therefore do not raise the same risks of double dipping.

Response:We sincerely thank the reviewer for raising this critical methodological point. We fully agree that using a data-driven cutoff visually identified from an outcome-associated RCS curve for subsequent hypothesis testing constitutes a form of “double-dipping” and can invalidate the statistical inference.

In direct response to this concern, we have revised our primary analysis to eliminate this bias and have performed an additional sensitivity analysis to ensure the robustness of our conclusions. The revisions are as follows:

(1) Revised Primary Analysis (Median-Based Cutoff):

We have changed the method for dichotomizing the AIP in our main joint analysis. Instead of the point visually identified from the RCS curve, we now use the sample median of AIP as the cutoff point. The median value in our study cohort is -0.24. This approach defines the exposure category based solely on the sample distribution, independent of the outcome variable, thereby completely avoiding the “double-dipping” issue.

We have updated the Methods section accordingly: “For the primary joint analysis, AIP was dichotomized into low and high groups based on the sample median value (-0.24). To test the robustness of our findings, we performed a sensitivity analysis in which AIP was dichotomized using a cut-off of -0.16, which has been validated in prior research (23)..” (Methods:Statistical analysis)

An important observation is that the sample median (-0.24) is numerically identical to the inflection point noted in our initial exploratory RCS curve. Therefore, while the methodological justification has been fundamentally strengthened, the dichotomization threshold and all numerical results presented in Table 3 remain unchanged. The estimates for the joint associations are preserved but are now methodologically robust.

(2) New Sensitivity Analysis (Literature-Informed Cutoff):

To further address the reviewer’s suggestion and demonstrate that our findings are not contingent on a specific cutoff choice, we conducted a sensitivity analysis using an alternative, clinically-informed AIP cutoff of -0.16, as referenced in prior literature. This cutoff was not derived from our dataset. The results of this analysis are presented in Table S5 (S6 Apeendix). As reported in the revised Results section, the pattern of association from this sensitivity analysis is fully consistent with our primary findings from Table 3.

“In sensitivity analysis using an AIP cut-off value of -0.16 (Table S5, S6 Appendix), the pattern of association remained consistent.” (Results:Joint analysis of AIP and menopausal status on prediabetes)

We believe these revisions directly and satisfactorily resolve the methodological issue raised, significantly strengthening the validity and credibility of our findings. We thank the reviewer for this insightful comment, which has improved the rigor of our work.

Yours sincerely,

Hongxia Zhuo

Institution and address: Hand Surgery Department, Union Hospital, Tongji Medical College, Huazhong University of Science and Technology, Wuhan, Hubei, P.R.China

Email: zhx2020zrj@163.com

Huiwu Han

Institution and address: Teaching and Research Section of Clinical Nursing, Xiangya Hospital of Central South University, Changsha, Hunan, P.R. China

Email: hanhw8888@csu.edu.cn

---

## [Decision Letter · Decision Letter 2]

27 Jan 2026

Does menopause influence the association between atherogenic index of plasma and prediabetes? A cross-sectional study in middle-aged Chinese women

PONE-D-25-48687R2

Dear Dr. Zhuo,

We’re pleased to inform you that your manuscript has been judged scientifically suitable for publication and will be formally accepted for publication once it meets all outstanding technical requirements.

Kind regards,

Qian Wu

Academic Editor

PLOS One

Additional Editor Comments (optional):

Reviewers' comments:

Reviewer's Responses to Questions

**Comments to the Author**

1. If the authors have adequately addressed your comments raised in a previous round of review and you feel that this manuscript is now acceptable for publication, you may indicate that here to bypass the “Comments to the Author” section, enter your conflict of interest statement in the “Confidential to Editor” section, and submit your "Accept" recommendation.

Reviewer #2: All comments have been addressed

Reviewer #5: All comments have been addressed

2. Is the manuscript technically sound, and do the data support the conclusions?

Reviewer #2: Yes

Reviewer #5: Yes

3. Has the statistical analysis been performed appropriately and rigorously?

Reviewer #2: Yes

Reviewer #5: Yes

4. Have the authors made all data underlying the findings in their manuscript fully available?

Reviewer #2: Yes

Reviewer #5: Yes

5. Is the manuscript presented in an intelligible fashion and written in standard English?

Reviewer #2: Yes

Reviewer #5: Yes

6. Review Comments to the Author

Reviewer #2: The deposition of data to the Mendeley Data repository satisfies the journal's transparency requirements.

The inclusion of the formal interaction test (P=0.973) and the subsequent revision of the language (shifting from "moderation" to "joint association") significantly improves the scientific accuracy of the paper. The authors responsibly acknowledge that while postmenopausal status does not statistically modify the AIP slope, the combination of postmenopausal status and high AIP identifies the highest-risk group, which is a clinically valid observation.

The addition of sensitivity analyses regarding the AIP cutoff and outcome definitions addresses the methodological concerns raised previously.

The manuscript is now technically sound, policy-compliant, and appropriately cautious in its conclusions. I have no further comments.

Reviewer #5: (No Response)

7. PLOS authors have the option to publish the peer review history of their article (what does this mean?). If published, this will include your full peer review and any attached files.

Reviewer #2: No

Reviewer #5: No

---

## [Editor Report · Acceptance letter]

PONE-D-25-48687R2

PLOS One

Dear Dr. Zhuo,

I'm pleased to inform you that your manuscript has been deemed suitable for publication in PLOS One. Congratulations! Your manuscript is now being handed over to our production team.

Kind regards,

on behalf of

Dr. Qian Wu

Academic Editor

PLOS One